# Systematic Identification of Familial Hypercholesterolaemia in Primary Care—A Systematic Review

**DOI:** 10.3390/jpm11040302

**Published:** 2021-04-15

**Authors:** Luisa Silva, Nadeem Qureshi, Hasidah Abdul-Hamid, Stephen Weng, Joe Kai, Jo Leonardi-Bee

**Affiliations:** 1Primary Care Stratified Medicine (PRISM) Group, NIHR School of Primary Care Research, University of Nottingham, Nottingham NG7 2RD, UK; luisa.silva@nottingham.ac.uk (L.S.); msxha27@exmail.nottingham.ac.uk (H.A.-H.); stephen.weng@evda.co.uk (S.W.); joe.kai@nottingham.ac.uk (J.K.); 2Department of Primary Care Medicine, Faculty of Medicine, Universiti Teknologi MARA, Sungai Buloh 47000, Malaysia; 3Centre for Evidence Based Healthcare, Division of Epidemiology and Public Health, School of Medicine, University of Nottingham, Nottingham NG7 2RD, UK; mczjl@exmail.nottingham.ac.uk

**Keywords:** familial hypercholesterolaemia, primary care, genetics

## Abstract

Familial hypercholesterolaemia (FH) is a common inherited cause of premature cardiovascular disease, but the majority of patients remain undiagnosed. The aim of this systematic review was to assess the effectiveness of interventions to systematically identify FH in primary care. No randomised, controlled studies were identified; however, three non-randomised intervention studies were eligible for inclusion. All three studies systematically identified FH using reminders (on-screen prompts) in electronic health records. There was insufficient evidence that providing comments on laboratory test results increased the identification of FH using the Dutch Lipid Clinic Network (DLCN) criteria. Similarly, using prompts combined with postal invitation demonstrated no significant increase in definite FH identification using Simon-Broome (SB) criteria; however, the identification of possible FH increased by 25.4% (CI 17.75 to 33.97%). Using on-screen prompts alone demonstrated a small increase of 0.05% (95% CI 0.03 to 0.07%) in identifying definite FH using SB criteria; however, when the intervention was combined with an outreach FH nurse assessment, the result was no significant increase in FH identification using a combination of SB and DLCN criteria. None of the included studies reported adverse effects associated with the interventions. Currently, there is insufficient evidence to determine which is the most effective method of systematically identifying FH in non-specialist settings.

## 1. Introduction

Familial hypercholesterolaemia (FH) is an autosomal-dominant disease and has long been recognized as a cause of premature coronary heart disease (CHD) [1]. It is associated with mutations in four genes: low-density lipoprotein (LDL) receptor, apolipoprotein B (Apo B), proprotein convertase subtilin/kexin 9 (PCSK9) and low-density lipoprotein receptor adaptor protein (LDLRAP) [2]. The majority of people with FH have the heterozygous form, with an estimated prevalence up to 1 in 200 [1,3]. In the most recent studies, compared to the general population, these patients have around a thirteen-fold increase in CHD mortality [1,4]. FH can also be inherited in an homozygous form, albeit much more rarely, with an estimated prevalence ranging from 1 in 160,000 to 1 in 1,000,000 individuals [2]. Further, a rarer autosomal-recessive form of the disease also exists [2]. Lipid-lowering treatment reduces CHD mortality by 44% [5]. However, up to 80% of individuals with FH remain undiagnosed and therefore untreated, resulting in major opportunities to prevent premature heart disease [1,3,6].

Several national guidelines on identifying FH have been published [7,8,9,10,11,12,13]. In these guidelines, confirmation of FH diagnosis involves assessment against one or more specified diagnostic criteria, such as the SimonBroome (SB) criteria [11], the USMedPed criteria [13], the Dutch Lipid Clinic Network (DLCN) criteria [14,15,16], and the Japanese criteria [8]. 

Currently, individuals with FH are found incidentally in usual practice. It has been suggested that a more systematic approach may help to identify more individuals in the non-specialist setting [15,17,18].

The objective of this review is to assess the effectiveness of systematically identifying FH in the adult primary care population, compared to usual care, to detect and manage this condition. 

## 2. Materials and Methods

This systematic review was conducted following the recommendations of the Cochrane Handbook for Systematic Reviews of Interventions [19] and reported according to the Preferred Reporting Items for Systematic Reviews and Meta-Analysis (PRISMA) statement [20]. A related Cochrane protocol using more stringent study design inclusion criteria is reported elsewhere [21]. 

### 2.1. Eligibility Criteria 

#### 2.1.1. Types of Study

We included randomised controlled trials (RCTs) and non-randomised intervention studies, including, but not limited to, controlled and uncontrolled before and after studies. We excluded non-randomised intervention studies, which did not report baseline data of usual care (defined as incidental and non-systematic care during routine consultation with participants [22]).

#### 2.1.2. Types of Participant

We included adult participants (over 17 years old) who accessed primary care. Participants with a previous diagnosis of FH or other inherited lipid disorders were excluded.

#### 2.1.3. Types of Intervention

We included any interventions that aimed to systematically identify people with possible or definite FH. Specialists delivering the interventions in primary care (e.g., FH nurse specialists) were eligible for inclusion. Eligible comparators were usual care, including incidental and non-systematic intervention during routine consultation with participants. This includes noting a raised cholesterol during consultation with individuals presenting with concerns about their personal or family history. 

#### 2.1.4. Types of Outcome Measure

##### Primary Outcomes: 

Diagnosis of definite FH (i.e., a positive genetic mutation or clinical characteristics of FH);Diagnosis of possible and probable FH (as defined by diagnostic criteria);Adverse events associated with the intervention.

##### Secondary Outcomes: 

Cholesterol levels;Cardiovascular mortality and morbidity (minimum of one-year follow-up);Lipid-lowering treatment;Referral to a specialist service;Adverse self-reported psychological effects.

### 2.2. Search Strategy 

We identified relevant studies from a comprehensive search of seven electronic databases, grey literature sources and Clinical Trials Registries. Additionally, we hand searched five major journals, two online resources from HEART UK [23] and the FH Foundation [24], and relevant guideline developers (National Institute for Health and Care Excellence [10], Scottish Intercollegiate Guidelines Network [25]) to identify further eligible studies (Appendix A for full details). 

### 2.3. Study Selection 

Two authors independently screened the titles and abstracts of the studies identified from the searches. The full texts of the potentially eligible studies were also screened independently by two different authors, and reasons for exclusion at the full text stage were noted. Any disagreements were resolved through discussion, or where necessary, with the assistance of a third author.

### 2.4. Data Extraction

Two authors independently extracted data from included studies using a previously piloted extraction form. The data extracted included author name, year of publication, characteristics of the population, intervention, and comparator. Discrepancies were resolved through discussion with a third author. Authors of the included studies were contacted where additional information was required.

### 2.5. Risk of Bias Assessment 

Two review authors independently assessed the risk of bias using the ROBINS-I tool for non-randomised intervention studies [26]. A third author resolved disagreements. 

### 2.6. Data Synthesis 

Due to the small number of studies included in the review and the methodological differences between them, this was limited to a narrative synthesis based on identifying patterns within the results of the studies. The results for comparison between intervention and comparator groups for continuous outcomes are reported as mean differences (MD) with 95% confidence intervals (CI). P values less than 0.05 were deemed statistically significant. 

## 3. Results

### 3.1. Study Selection 

The search identified a total of 4638 citations, of which 32 citations were identified as potentially eligible based on title and abstract screening, representing 29 studies. Following full-text screening, three studies were eligible for inclusion into the review [27,28,29] (Figure 1: PRISMA diagram). Thus, 25 studies were excluded from the review at the full text stage. The reasons for exclusion were no baseline data of usual care (17 studies [30,31,32,33,34,35,36,37,38,39,40,41,42,43,44,45,46]), ineligible participants (7 studies [47,48,49,50,51,52,53]) and ineligible condition (1 study [54]). There was also one ongoing study [55]. Details on the 17 studies excluded based on study design (no baseline data of usual care) are included as Appendix A. 

### 3.2. Characteristics of Included Studies 

The three included studies were published between 2013 and 2018 (basic details in Table 1, with full details in Appendix A). Two studies [28,29] were carried out in the United Kingdom, and the remaining study [27] was conducted in Australia. All three primary care studies used an uncontrolled before-and-after study design. A total of 281,869 participants were involved in the studies, although the vast majority of participants were from one study [28]. The other two studies included a relatively small number of participants: 96 [27] and 118 [29].

The interventions assessed within the three included studies focused on systematically identifying participants through electronic health record (EHR) searches and using computer reminders; however, the modalities varied between studies. In two studies, the computer reminders were on-screen prompts for participants who had raised cholesterol recorded within their EHR [28,29]. The Weng study [29] also incorporated postal invitations to be assessed for FH by completing a family history questionnaire. Additionally, in the follow-up phase of the Green study [28], a combined intervention of on-screen prompts with an invitation for clinical assessment by a specialist FH nurse was also assessed. In the third study, the computer reminder was an on-screen prompt in the downloaded laboratory results [27]. 

The three studies made a diagnosis of definite, probable and possible FH based on either the Simon-Broome criteria [28,29] or a modified version of the DLCN criteria (where only participants with LDL-C ≥ 6.5 mmol/L were selected) [27]. However, in the follow-up phase of the Green study [28], a diagnosis of FH was based on using a combination of the Simon-Broome and/or DLCN criteria.

The overall risk of bias for all three of the included studies was low [27,29], including the EHR reminder phase in Green [28]; however, for the combined intervention of computer on-screen prompts and FH nurse intervention in the follow-up phase of the Green study [28], the overall risk of bias was moderate to account for the risk of attrition bias due to missing outcome data for some participants (data could not be extracted for 62,684 out of 262,030 participants) (Appendix A). 

#### 3.2.1. Diagnosis of Definite FH

All included studies reported the number of participants identified with definite FH. In the Green study, systematically identifying participants in the EHRs using on-screen prompts found a small absolute improvement in the proportion of participants diagnosed with FH with the prevalence increasing from 0.13% to 0.18% over the two years of the intervention period (SB criteria: MD 0.05%, 95% CI 0.03 to 0.07%) [28]. Additionally, during the follow-up phase of this study, combining on-screen prompts in the EHR with an invitation to an assessment with a specialist FH nurse, led to a further marginal increase in prevalence of definite FH to 0.19% (SB criteria: MD 0.07%, 95% CI 0.05 to 0.09%). 

In the Weng study, it was found that systematically identifying participants in EHRs using on-screen prompts and postal invitations to complete family history questionnaires may slightly increase the number of patients diagnosed with definite FH (SB criteria: 0 at baseline versus 2 diagnoses 6 months post-intervention; MD 1.69%, 95% CI –1.69% to 5.97%) [29]. 

Similarly, in the Bell study, it was found that systematically identifying participants in EHRs using on-screen prompts within the laboratory results may slightly increase the number of diagnoses with definite FH (modified DLCN criteria: 0 versus 2 diagnoses 4 months post-intervention; MD 2.08%, 95% CI –2.05% to 7.28%) [27]. Both diagnoses in this study had an identifiable LDL-receptor gene mutation on genetic testing. 

#### 3.2.2. Diagnosis of Possible and Probable FH

Systematically identifying participants in EHRs, using on-screen prompts, found a small absolute improvement in the proportion of participants diagnosed with possible FH from the baseline (SB criteria: MD 0.04%, 95% CI 0.03% to 0.05%) [28]. Additionally, combining on-screen prompts in the EHR with an invitation for a specialist FH nurse assessment had an added small absolute improvement in the diagnosis of possible FH from the baseline (SB criteria: MD 0.02%, 95% CI 0.02% to 0.03%) [28]. 

In contrast, using on-screen prompts in EHRs, combined with postal invitations to complete family history questionnaires, resulted in a 25% absolute improvement in the identification of possible FH (SB criteria: 0 at baseline versus 30 cases post-intervention; MD 25.42%, 95% CI 17.75% to 33.97%) [29].

However, systematically identifying participants in EHRs using on-screen prompts in the laboratory results did not increase the diagnosis of probable FH (modified DLCN criteria: MD 2.08%, 95% CI –2.05% to 7.28%) [27].

#### 3.2.3. Adverse Events Associated with the Intervention

None of the included studies reported adverse events.

#### 3.2.4. Cholesterol Levels

In the Weng study, there was no significant reduction in total cholesterol in the intervention group compared to the baseline (total cholesterol: MD –0.16, 95% CI –0.78 to 0.46) [29]. There were mixed results for LDL cholesterol in the Weng study, with no significant changes (LDL-C: MD –0.12, 95% CI –0.81 to 0.57) [29]. However, the Bell study found a significant decrease 12 months after intervention (LDL-C: MD –3.00, 95% CI –3.42 to –2.58) [27]. 

#### 3.2.5. Cardiovascular Mortality and Morbidity

None of the included studies reported this outcome measure.

#### 3.2.6. Lipid-Lowering Treatment

Using on-screen prompts in EHRs, combined with postal invitations to complete family history questionnaires, resulted in an absolute increase of 19% in the prescribing of statins (MD 18.75%, 95% CI 8.9% to 35.3%), and an absolute increase of 9% in the prescribing of high potency statins (MD 9.38%, 95% CI 3.2% to 24.2%), compared to the baseline [29].

#### 3.2.7. Referral to a Specialist Service

One study reported that systematically identifying participants in EHRs using on-screen prompts in the laboratory result resulted in 4% (4/96) of participants being referred to a specialist service; however, at baseline, outcome data were not reported [27]. 

#### 3.2.8. Adverse Self-Reported Psychological Effects

None of the included studies reported this outcome measure.

## 4. Discussion

This is the first systematic review to assess the effectiveness of interventions to systematically identify FH in primary care. The review included three studies, comprising a total of 281,869 participants; however, the vast majority of participants were from one study [28]. The included studies used on-screen prompts in electronic health records to systematically identify individuals with raised cholesterol; one study combined this with postal invitations to complete family history questionnaires [29] and another study combined the intervention with the invitation for clinical assessment by an FH specialist nurse [28]. A small absolute improvement in the identification of individuals with definite FH was seen in one study [28]; however, no significant evidence of an effect was seen in the other two studies. Given that FH is a relatively rare condition and studies were undertaken in a few general practices, it is not surprising that the absolute improvement in detecting the condition is quite small. 

As the target populations within the included studies varied, it was difficult to establish direct comparisons between the three studies. Green [28] assessed the whole practice population, whilst Bell [27] and Weng [29] had a pre-selected sample based on specific case-finding criteria. 

Improvements in the detection of possible FH with on-screen prompts varied between studies. One study, when combined with postal invitations to complete a family history questionnaire, demonstrated a larger absolute improvement [29], whilst another studies’ prompts, with or without FH specialist nurse assessment, found small improvement [28]. However, using on-screen prompts in downloaded laboratory results demonstrated no improvement [27].

Further, on-screen prompts had no significant effect on total cholesterol and only one study showed a significant decrease in LDL-cholesterol levels [27]; an increase in the prescribing of statins and high potency statins was demonstrated after introducing on-screen prompts with postal invitations to complete family history questionnaires [29]. 

The number of participants clinically diagnosed with FH was relatively small across the three included studies, which makes it difficult to ascertain the effectiveness of the systematic identification as there will be limited power within the analyses. Furthermore, several of the pre-specified primary and secondary outcome measures were not assessed within the studies and, additionally, the length of follow-ups was too short to evaluate cardiovascular disease outcome measures.

As one of the included studies had been undertaken by some of the review authors, to avoid bias, two independent reviewers extracted the data and assessed for risk of bias. Two of the included studies were deemed to have a low overall risk of bias; however, the inherent biases associated with the design of non-randomised intervention studies means that they are not optimal for assessing the effectiveness of the intervention. 

There have been no previous published systematic reviews on this topic. However, there have been several narrative reviews on the detection of FH in primary care. For example, Lan et al. [56] outlines different screening methods for FH, including primary care Electronic Health Records searches. 

The updated (English) NICE guidelines on FH identification suggest that clinicians should systematically search primary care Electronic Health Records for individuals at risk of FH [10]. One of the studies [28] included in the current review also informed the recommendations in the NICE guideline. Other studies listed in the updated NICE guideline as evidence for the use of Primary Care Electronic Health Records were also part of our search results. However, these studies were excluded from the current systematic review as they did not provide baseline data on usual care to allow an assessment of improvement in outcome measures [32,38,40,41,44]. 

Further high-quality, controlled studies, ideally in the form of randomised, controlled trials, are needed before a firm conclusion regarding the most effective approach to systematically identifying individuals at risk of FH. Furthermore, to fully address this research question, studies should use recognised clinical assessment criteria for the diagnosis of FH and, if possible, genetic confirmation of FH. Future studies should provide detailed descriptions of the relevant process and outcome measures, including detection of genetically confirmed FH and both surrogate and clinical outcome measures of cardiovascular disease.

The review provided insufficient evidence to recommend specific systematic approaches to identify individuals at increased risk of FH. However, this does indicate the clinical value of searching primary care Electronic Health Records (EHR). The simplest iteration of this is searching the records for individuals with raised cholesterol, and more elaborate methods of searching primary care EHR are being developed [57]. Further, automated alerts to identify patients at increased risk, based on cholesterol levels, have been developed. The latter approach is already a recognized strategy in primary care; for example, haematology reports alerting clinicians to screen for thalassaemia traits when a microcytic, hypochromic blood profile is identified. 

## 5. Conclusions

This systematic review provided insufficient evidence to inform the most effective approach to identify FH in primary care. There is a potential role for searching primary care Electronic Health Records, but further studies need to be conducted. 

## Figures and Tables

**Figure 1 jpm-11-00302-f001:**
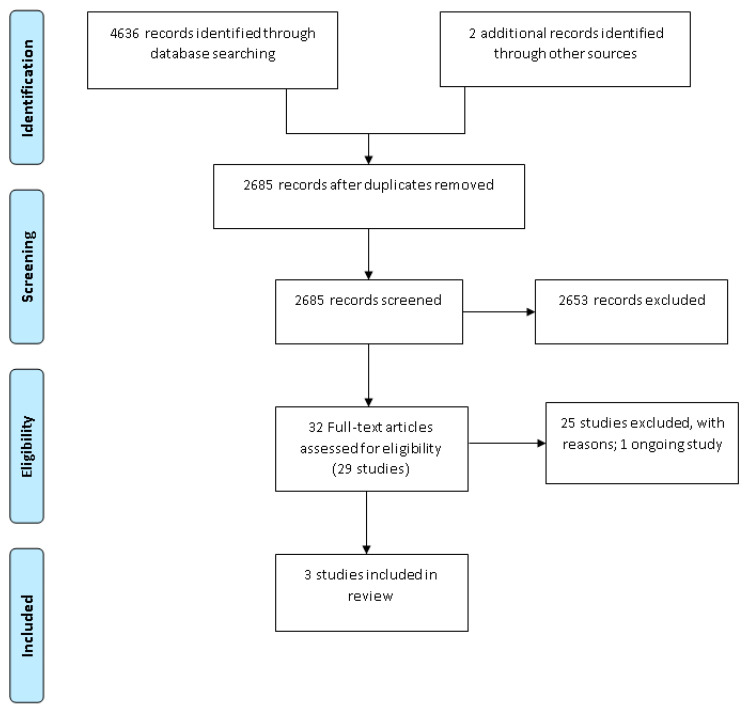
Preferred Reporting Items for Systematic Reviews and Meta-Analysis (PRISMA) diagram.

**Table 1 jpm-11-00302-t001:** Summary of included studies.

Study and Year	Design/ Setting	Participants	Intervention	Outcomes	Comparisons	Main Results		
						Pre-Intervention	Post-Intervention	Absolute Difference (95% CI), n
Bell et al., 2013	Uncontrolled BA study General practices in Western Australia	96 Patients Gender: Female 68 (70.9%), Male 28 (29.1%) Age (years): mean ± SD [range]: 53.7 ± 10.7 [25,26,27,28,29,30,31,32,33,34,35,36,37,38,39,40,41,42,43,44,45,46,47,48,49,50,51,52,53,54,55,56,57]	Interpretative comments added to lipid results	-FH diagnosis (Modified DLCN criteria) -LDL-c level -Referral to specialist	No comments added to lipid results (standard/usual care)	Definite FH: 0/96 (0%) Possible FH: 0/96 (0%)	Definite FH: 2/96 (2.08%) Possible FH: 2/96 (2.08%)	Definite FH: 2.08% (–2.05 to 7.28%), *n* = 96 Possible FH: 2.08% (–2.05 to 7.28%), *n* = 96
Green et al., 2016	Uncontrolled BA study with two sequential interventions. General practices in South East England	Approximately 290,000 patients Gender: not reported Age: In 2011, 37,200 people were aged >65 years and 4400 aged >85 years	1: computer based reminder message 2: FH Nurse Advisor Programme—2-part process. Part 1 as above; Part 2 involved consultation with nurse to collect further information	-FH diagnosis (Baseline: S-B; Post-interv: S-B and/or DLCN criteria)	Baseline prevalence of FH	EHR Search and reminder		
						Definite FH: 331/262,030 (0.13%) Possible FH: 12/262,030 (0.005%)	Definite FH 354/199,346 (0.18%) Possible FH: 88/199,346 (0.04%)	Definite FH: 0.05% (0.03 to 0.07%), *n* = 262,030 Possible FH: 0.04% (–0.03 to 0.05%), *n* = 262,030
						EHR search and reminder + nurse intervention		
						Definite FH: 331/262,030 (0.13%) Possible FH: 12/262,030 (0.005%)	Definite FH: 546/281,655 (0.19%) Possible FH: 147/281,655 (0.05%)	Definite FH: 0.07% (0.05 to 0.09%), *n* = 262,030 Possible FH: 0.05% (0.04 to 0.06%), *n* = 262,030
Weng et al., 2018	Uncontrolled BA study Six General Practices in Central England	831 identified, 118 patients medical records accessed Gender: Female 46 (39%), Male 72 (61%) Age (years) mean (SD): male 58 (9.0), female 56 (7.5)	Combined approach: Opportunistic recruitment following computer-based reminder message with systematic postal recruitment of eligible patients	-FH diagnosis (S-B criteria) -Cholesterol -Statins prescribed	Same 118 participants with Cholesterol ≥ 7.5 mmol/L after the release of the NICE FH guidelines	Definite FH: 0/118 (0%) Possible FH: 0/118 (0%)	Definite FH: 2/118 (1.69%) Possible FH: 30/118 (25.42%)	Definite FH: 1.69% (–1.69 to 5.97%), *n* = 118 Possible FH: 25.42% (17.75 to 33.97%), *n* = 118

Abbreviations: CI = confidence interval; BA = before-and-after; DLCN = Dutch Lipid Clinic Network; FH = Familial Hypercholesterholaemia; SB = Simon-Broome; EHR = Electronic Health Records.

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
