# Peer review of "Systematic Identification of Familial Hypercholesterolaemia in Primary Care—A Systematic Review"

_jpm, 2021, doi:10.3390/jpm11040302_

Round 1
Reviewer 1 Report
In the manuscript “Systematic identification of familial hypercholesterolaemia in 2 primary care – a systematic review”, the author aimed to summarize the effectiveness of interventions to systematically identify FH in primary care.
The topic of the manuscript is certainly interesting and captures one of the issues of the moment on cardiovascular prevention, in light of new scientific evidence on the topic.
However, there are some points that need further clarification:
- In the introduction, the authors state, " Familial hypercholesterolaemia (FH) is an autosomal-dominant disease…". Please specify that familial hypercholesterolemia is in most cases autosomal dominant, but autosomal recessive forms also exist.Furthermore, no no mention of the epidemiology of the homozygous form
- In the introduction section, specify the most frequent genetic mutations that cause familial hypercholesterolemia, as reported in the following references:
- Clinical Genetic Testing for Familial Hypercholesterolemia: JACC Scientific Expert Panel - https://doi.org/10.1016/j.jacc.2018.05.044
- Beyond cholesterol metabolism: The pleiotropic effects of proprotein convertase subtilisin/kexin type 9 (PCSK9). Genetics, mutations, expression, and perspective for long-term inhibition. BioFactors. 2020;1–14. https://doi.org/10.1002/biof.1619
- Familial hypercholesterolemia--epidemiology, diagnosis, and screening - Curr Atheroscler Rep. 2015;17(2):482.doi: 10.1007/s11883-014-0482-5
- A "conclusion" section is missing. Please provide.
- Please provide additional insights into the discussion, of the usefulness of the review findings. How can it impact clinical practice?
- Please, you should explain each of your abbreviations the first time it appears in the main text (es. SB – Simon-Broome, etc) and provide a description of the abbreviations for Table 1 in the caption.
Author Response
We thank the reviewer for identifying this article would be of interest to your readership as it relates to cardiovascular disease prevention.
Considering their specific queries:
(1) In the introduction, the authors state, " Familial hypercholesterolaemia (FH) is an autosomal-dominant disease…". Please specify that familial hypercholesterolemia is in most cases autosomal dominant, but autosomal recessive forms also exist. Furthermore, no mention of the epidemiology of the homozygous form
Thank you for your advice on providing further information about FH. We have added on line 37 “FH can also be inherited in an homozygous form, albeit much more rarely, with an estimated prevalence ranging from 1 in 160.000 to 1 in 1.000.000 individuals [2].”
Further on line 39, we have added “Further, a rarer autosomal-recessive form of the disease also exists [2].”
(2) In the introduction section, specify the most frequent genetic mutations that cause familial hypercholesterolemia, as reported in the following references:
We thank the reviewer for suggested references. As suggested we have added more information on genetic mutations, citing one of the references, on line 31: “It is associated with mutations in four genes: LDL receptor, apolipoprotein B (Apo B), proprotein convertase subtilin/kexin 9 (PCSK9) and low-density lipoprotein receptor adaptor protein (LDLRAP) [2]. “
(3) A "conclusion" section is missing. Please provide.
We have added a conclusion section in line 288 – “This systematic review provided insufficient evidence to inform the most effective approach to identify FH in primary care. However, there is a potential role for searching primary care Electronic Health Records but further studies need to be conducted”
(4) Please provide additional insights into the discussion, of the usefulness of the review findings. How can it impact clinical practice?
Following your suggestion, we have expanded the discussion section – please see line 279 – “The review provided insufficient evidence to recommend specific systematic approaches to identify individuals at increased risk of FH. However, this does indicate the clinical value of searching primary care Electronic Health Records (EHR). At the simplest level, is searching the records for individuals with raised cholesterol. More elaborate methods of searching primary care EHR are being developed [58]. Further, automated alerts to identify patients at increased risk, based on cholesterol levels, have been developed. The latter approach is already a recognized strategy in primary care, for example, haematology reports alerting clinicians to screen for thalassaemia trait when a microcytic, hypochromic blood profile is identified”
(5) Please, you should explain each of your abbreviations the first time it appears in the main text (es. SB – Simon-Broome, etc) and provide a description of the abbreviations for Table 1 in the caption.
Thank you for pointing this out, we have clarified abbreviation for Simon-Broome (SB) in line 45 and also added description of abbreviations for Table 1 – line 143.
Reviewer 2 Report
Familial hypercholesterolemia is one of the most common monogenic diseases. Despite the presence of specific criteria for the diagnosis of FH and effective lipid-lowering drugs (statins, ezetimibe and PCSK9 inhibitors), patients with FH are usually poorly diagnosed and do not reach the LDL-С goal, therefore, they have an early development of coronary artery disease and early mortality. The introduction of approaches to the early detection of FH patients in primary care (such as automatic analysis of electronic medical records, etc.) can improve the diagnosis and treatment of FH patients. Reviewing such methods and comparing their effectiveness is important and relevant. The authors provided a systematic review of this topic. Systematic review results are a bit disappointing. Most of the studies on this topic were excluded from the review and only 3 studies were retained (one with a large number of patients and two with a very small number). It seems to me that the result obtained is not representative, and its additional value is small. Perhaps the value of the review can be increased by including information from excluded studies.
Author Response
We thank the reviewer for their observation that such a systematic review of the topic is important and relevant. Considering the reviewer’s comment “The authors provided a systematic review of this topic. Systematic review results are a bit disappointing. Most of the studies on this topic were excluded from the review and only 3 studies were retained (one with a large number of patients and two with a very small number). It seems to me that the result obtained is not representative, and its additional value is small. Perhaps the value of the review can be increased by including information from excluded studies.”
The systematic review used a comprehensive and robust search and selection criteria. To allow the evaluation of the intervention, we included all RCT and Non-randomised Studies of Interventions (NRSI) study designs which had a comparator group – for NRSI designs, we further allowed for the comparator group to either be a separate control group or the comparative baseline data before the intervention was implemented. However, please note that even this level of evidence was considered too weak by Cochrane collaboration to be included in a Cochrane systematic reviews, where they only include RCTs or controlled NRSI study designs. Similar to Reviewer 2, we would have anticipated that more studies would have met our relaxed inclusion criteria, but only 3 papers were found to be eligible. Whilst it is common practice for only the reasons for exclusion to be presented for studies which were excluded at the full text stage, we recognise the value of including some details of the interventions considered in the studies which did not have comparative baseline data and have therefore included this information in a supplementary table. Hence, the readership is signposted on line 130 in “3.1. Study selection” to the supplementary table.
Round 2
Reviewer 2 Report
I agree with all the edits, the article can be accepted.